# Rethinking Industrial Anomaly Detection in the Era of Large Vision-Language Models

## Abstract

State-of-the-art methods for industrial anomaly detection (IAD) typically rely on a training set of images to define normal conditions, flagging any deviations as anomalies. Obtaining this training set has two main issues - it is time consuming to obtain an extensive labeled set, and the assumption that all patterns outside the training set are truly anomalous is often unrealistic. Many rare patterns not captured in the training set, such as environmental changes, positional changes, or permissible deformation, may not constitute actual industrial defects. In this paper, we reframe the IAD task by using large vision-language models (LVLMs) without fine-tuning on training images. LVLMs can interpret and generalize from a single reference image, and can be more robust to rare but acceptable changes in images. Our experiments on two popular benchmarks, MvTec-AD and VisA, show that LVLMs with just one image and a textual description is competitive with state-of-the-art models, and offer a more robust and generalizable solution even with variations in testing images. We also identify a key limitation: LVLM performance degrades when detecting small anomalies. Despite this, our findings highlight the potential of LVLMs as a flexible and scalable foundation for industrial anomaly detection, opening new directions for future research.

## 1 Introduction

The recent development of deep learning models for industrial anomaly detection (IAD) opened up more opportunity for higher standards of manufacturing, with less discarded items and lower costs (Liu et al., 2024). State-of-the-art (SOTA) approaches to solving IAD using deep learning models have been diverse and extensive, and, in many cases, report high performance (Liu et al., 2024). These methods typically rely on large datasets of images depicting normal operating conditions. However, it is difficult to obtain images that fully cover all variations of normal conditions. Uncommon characteristics of images that are not included in the training set may be considered an anomaly, even when the image is normal.

While SOTA IAD systems are developed using training images, we propose an alternative approach to use large vision-language models (LVLMs) that have superior generalizing capabilities. Recent advancements in LVLMs have shown promising performance on wide ranges of tasks such as reasoning, question answering, and image captioning (Zhang et al., 2024a). We hypothesize that LVLMs can show IAD capabilities without any fine-tuning, making them a flexible and robust alternative to new IAD tasks. Because LVLMs have generalization capabilities through their extensive training with language and vision, LVLMs can bypass the need to obtain a large training dataset for every new IAD task. This minimizes the time and labor needed to obtain and label training images. This also allows one LVLM model to be deployed for many different tasks of IAD, which minimizes the time needed to build specialized models.

In this paper, we assess whether LVLMs, prompted with a single reference image and a generic prompt, can match or exceed the performance of specialized models, particularly in settings where anomalies are diverse or visually subtle. We evaluate using a simple method with LVLM on two standard IAD datasets with modified test images, and compare the results against SOTA models for IAD. In Section 2, we discuss relevant models and methods that have been used for IAD. In Section 3, we discuss the justification for and details of using LVLM for IAD. In Section 4, we discuss the datasets used, details of SOTA models selected

for comparison, and modifying the images to reflect a more realistic setting. Lastly, in Section 5, we discuss the results on the original datasets and modified images.

## 2   Related Work: Defining Anomaly with Images

Existing approaches to IAD have various approaches. Reconstruction based approaches use a model trained on normal images to compare the new reconstructed image with the original image, and use the error as a measure of anomaly. Some methods include AMI-Net, Dinomaly, and RealNet (Luo et al., 2024) (Guo et al., 2025) (Zhang et al., 2024b). Embedding based approaches are another solution where the images are converted to vectors using a pretrained model to determine an anomaly score. Some methods include Simple-FPN and ReConPatch (Zhao et al., 2024) (Hyun et al., 2024). Both reconstruction and embedding approaches rely on training a model using normal images, which can lead to two issues. First, obtaining a large training dataset can be difficult and time consuming. Second, these normal images must cover all variations of normal, since a rare normal image that is not in this training set can be labeled as an anomaly.

IAD approaches can also be categorized as single and multi-class methods. Single class based methods such as PaDiM are trained only on one object class and have a unique model for each object (Defard et al., 2020). Multi-class methods such as UniAD are able to detect anomalies for different object classes under a unified framework (You et al., 2022). While multi-class methods offer more flexibility, these methods are still trained on all objects, and therefore do not eliminate the time needed to obtain and label a training dataset. These models also use the anomaly mask labels to determine the location of the anomaly to train these models. Obtaining these anomaly mask labels can be even more time consuming.

## 3   Evaluating IAD in the LVLM Era

In contrast to previous methods, our approach using LVLM models offer an alternative solution to IAD. A training set is not necessary for LVLM models with a one-shot approach, which only uses a single normal reference image. One model can be used for various object classes. Therefore, LVLMs offer a multi-class, no training alternative to IAD. In this section, we first discuss the difficulty of defining an anomaly task with training images. Then, we introduce our simple LVLM method.

### 3.1   Defining anomaly is incomplete with images

The main issue with the standard IAD datasets is that it only uses images to define a IAD task. In practice, there may be other forms of normal conditions that may be missed in the training dataset. We had the opportunity to discuss an IAD task with Anonymous Corporation to detect anomalies at two key points of the manufacturing process, shown in Figures 1 and 2. Figure 1 shows a split pin, which is defined as normal if it is slotted correctly through the center nut and its two legs are bent at at least 120 degrees apart from each other, and finished with a paint mark on the bend. However, as demonstrated by Figures 1(a) and (b), there are still significant differences between the normal images. The object is free to rotate, the camera position may shift, and other objects may be present such as the 2 or 4 screws around the center nut. In Figure 1(c), the split pin is shown to be an anomaly because the legs have not been bent, but other differences such as the arm or blur, are not relevant to the IAD task. Similarly, Figure 2 shows a circular pin, which must be correctly placed into the groove such that the two holes at the end of the ring are close together and painted over. However, both normal and anomaly images show differences such as shadows and orientations that are not relevant to the IAD task. Covering all possible variations of normal images in the training set is not possible, which can lead to missed detection in practice, and only creates an argument for using more human labor and time to obtain additional training images.

### 3.2   Detecting anomaly with LVLMs

We hypothesize that using using an LVLM can help in defining an IAD task more clearly without needing a large training dataset. Recent developments in LVLMs have shown promising and powerful capabilities in image and text comprehension (Zhu et al., 2023). The multi-image capabilities of few-shot prompting, when

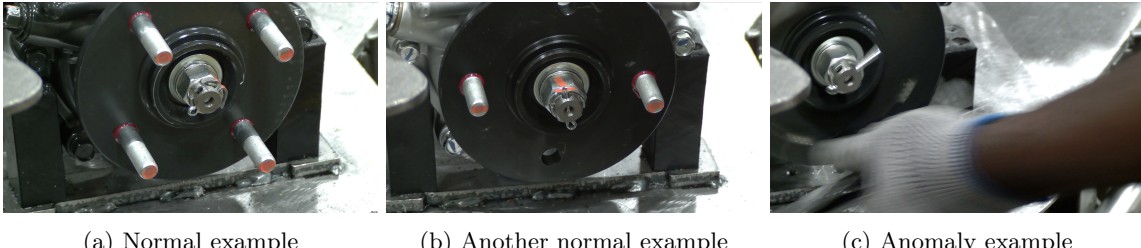

(a) Normal example        (b) Another normal example        (c) Anomaly example

Figure 1: Examples of normal and anomaly images of a split pin assembly obtained from Anonymous Corporation.

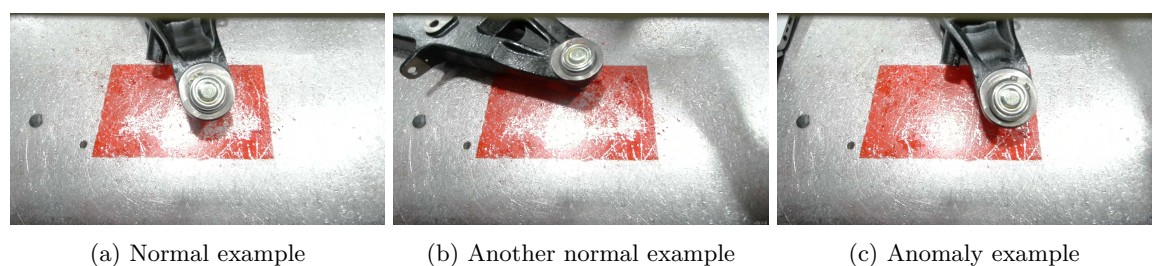

(a) Normal example        (b) Another normal example        (c) Anomaly example

Figure 2: Examples of normal and anomaly images of a circular pin assembly obtained from Anonymous Corporation.

combined with high performance of question-answering benchmarks, extends their conversational capabilities to understanding more complex contexts and tasks (Alayrac et al., 2022). One high performing LVLM is Qwen2-VL, an open source model that achieves impressive results comparable to leading models such as GPT-4o and Claude3.5-Sonnet across many multimodal benchmarks (Wang et al., 2024b). Based on its demonstrated performance, we use Qwen2-VL-72B-Instruct-AWQ model to evaluate IAD capabilities of LVLMs.

We introduce a simple, yet effective, method for prompting Qwen to evaluate its performance on the IAD datasets. We only use a single reference image such that we minimize the labor and time needed to develop a labeled training and testing dataset. The selected reference image is the first available image from the training set. We also use the same anomaly prompt on all image categories and datasets: "The second image is for anomaly detection. Does the second image have any anomaly compared to the first image? Reply yes or no, then explain." For a few object categories, we use additional prompts to describe any variation in the training set that is not reflected in the reference image. For example, the toothbrush reference image has blue bristles, but other images in the training set contains red or yellow bristles. The toothbrush prompt is therefore: "The second image is for anomaly detection. Does the second image have any anomaly compared to the first image? Reply yes or no, then explain. Ignore the color of the bristles for an anomaly." The list of specific prompts used are shown in Appendix A.2. Many image classes do not need any specific prompts, making this approach simple and general. Figure 3 shows the method overview and the anomaly prompt text.

## 4  Experimental Setup

We first introduce the two widely used datasets for IAD. Next, we select three recently developed SOTA models for comparison against the LVLM performance. We discuss augmenting the datasets to include variations to these images that may occur naturally in practice. Lastly, we discuss an evaluation metric for comparing the three models against the LVLM method.

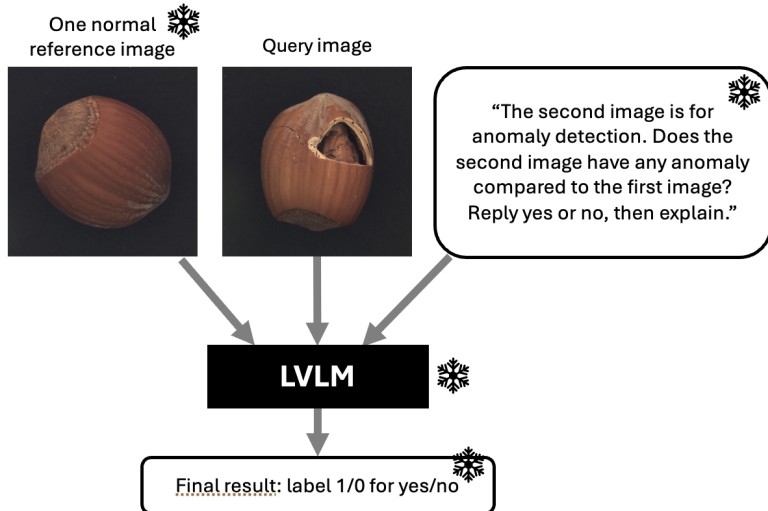

Figure 3: The LVLM simple method diagram, showing example images of the hazelnut in MvTec-AD dataset. In this method, the reference image, LVLM, and post processing functions are all frozen.

## 4.1 Datasets

The MvTec-AD dataset consists of 15 different image categories with 3629 normal images in the training set, and 1725 testing images with 1258 anomaly and 467 normal images (Bergmann et al., 2019). Four image categories are texture based, and the other 11 image categories are object based. All anomaly images are provided a ground truth mask to indicate the location of the anomaly. The size of the anomaly in the images relative to the total image range from 0.037% to 49.3% with an average anomaly size of 4.4%.

We repeated the experiments on the VisA dataset which contains 12 image categories for IAD (Zou et al., 2022). There are 8659 normal images in the training set and 2162 images in the testing set with 1200 anomaly and 962 normal images. These anomaly images are provided a ground truth mask to indicate the location of the anomaly. The size of the anomaly in the images relative to the total image range from 0.002% to 32.0%, with an average anomaly size of 1.0%.

## 4.2 State-of-the-art Models

We choose to compare the LVM approach with three SOTA models that achieve high performance on the MVTec-AD dataset: GLASS, INP-Former, and AnomalyGPT. (Chen et al., 2024)(Luo et al., 2025)(Gu et al., 2023). These models were chosen for their recent accomplishments in high IAD performance and the variety in solutions.

### 4.2.1 GLASS

Global and Local Anomaly co-Synthesis Strategy (GLASS) is chosen to compare against the LVLM strategy for IAD (Chen et al., 2024). This model uses a unified framework for analyzing a broad range of anomalies at image and pixel levels. Training this model is split into three branches - normal, GAS, and LAS. First, the normal training images are processed to obtain normal features. Then, these features are used as input to both the GAS and LAS branch. In the GAS branch, the global anomaly features are synthesized using gradient guidance. In the LAS branch, local anomaly features are synthesized by overlaying textures from the DTD dataset images (Cimpoi et al., 2014). These three features from each branch is fed into a trainable discriminator. In inference, only the normal branch is used to calculate an anomaly score for each image. This method is trained on normal images. This model was replicated using the public repository on Github.

The model weights for the MvTec-AD dataset were downloaded from the repository, while the model weights for the VisA dataset were trained using the published code and instructions.

### 4.2.2 INP-Former

The second model chosen to compare against the LVLM strategy for IAD is the Intrinsic Normal Prototypes - Former (INP-Former) (Luo et al., 2025). This method extracts INPs from the image by linearly combining normal tokens using the INP Extractor. The INP is used in the INP-Guided Decoder to reconstruct the normal tokens and uses the reconstruction errors between the original image as the anomaly score. The multi-class model weights were used in evaluation for our experiments, as made available in the public Github repository.

### 4.2.3 AnomalyGPT

The last model we used to compare is the AnomalyGPT model (Gu et al., 2023). This model is based on LVLM, and uses simulated anomaly images for training data. An image encoder and decoder are used to process the query image, then compared with normal and anomaly texts to obtain localization results. This is passed to the prompt learner to convert into prompt embeddings used as input to the LLM. ImageBind-Huge is used as the image encoder and Vicuna-7B is used as the LLM (Girdhar et al., 2023)(Chiang et al., 2023). It is claimed in this paper that this is the first attempt to use LVLMs in IAD. The supervised trained models published on the Github repository are used for our experiments.

### 4.3 Augmented testset: more variations to normal images

Although the academic datasets used in our experiments are widely used for research, the consistency of these images can be unrealistic. As discussed in Section 3, it can be difficult to take images from consistent conditions and angles, and variations in images are inevitable. We hypothesize that since LVLMs are not fine-tuned on a training set for specific IAD tasks, the performance of LVLMs should be more robust to these changes than traditional models. Both the reference image and the prompt is important in communicating the anomaly task, while keeping the task general enough to adapt to small changes during testing.

To explore the performance of SOTA models under these realistic conditions, we augmented the test dataset with modifications that may occur in an applied setting and do not affect the visibility of anomalies. Figure 4 and 5 show the original image and the modified images for the two datasets. Rotation modification rotates the image 90 degrees counterclockwise. Padding adds a white border around the image at 20 pixels thick. Lastly, a timestamp is added to the top right corner of the image saying "2025-05-12" in white text with black border in font size 30. Observing the performance on the original images and then comparing the performance on these modified images can demonstrate the robustness of the model to perturbations to the test image that were not shown in the training dataset.

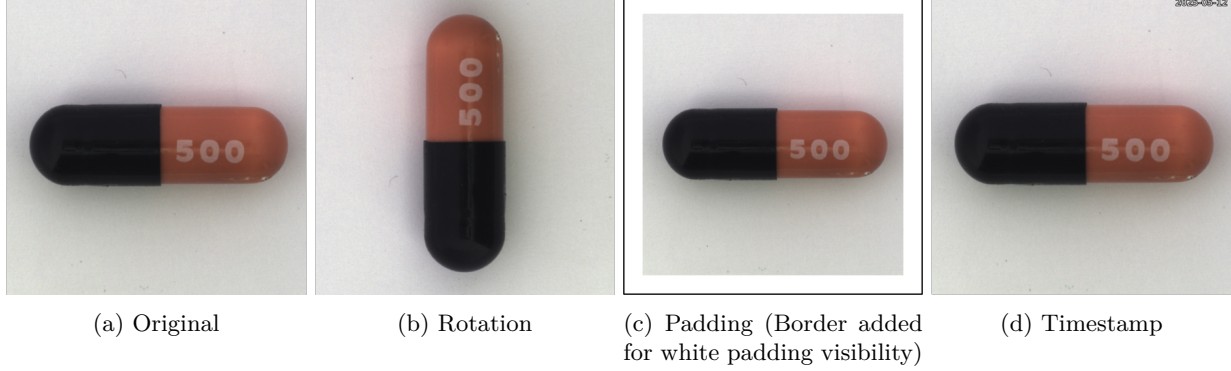

(a) Original     (b) Rotation     (c) Padding (Border added for white padding visibility)     (d) Timestamp

Figure 4: Examples of the original and modified images of a capsule in the MvTec-AD dataset.

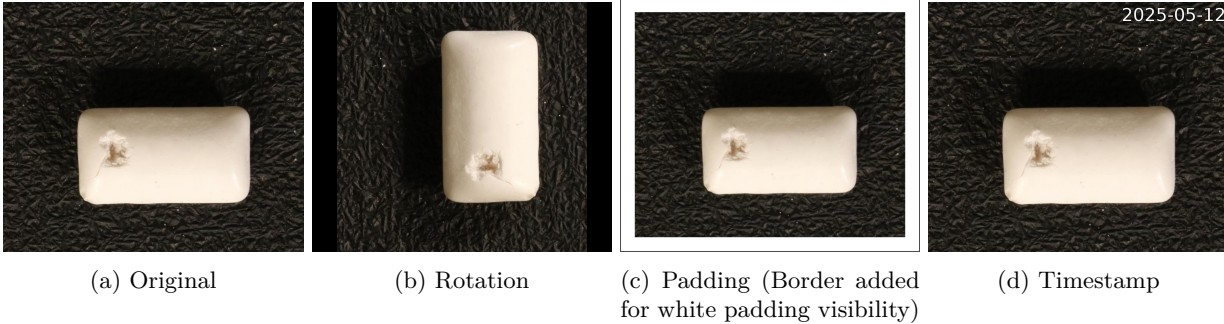

(a) Original      (b) Rotation      (c) Padding (Border added for white padding visibility)      (d) Timestamp

Figure 5: Examples of the original and modified images of a chewing gum in the VisA dataset.

## 4.4 Evaluation Metrics

The SOTA models standard of evaluation metric is Area Under the Receiving Operating Characteristic curve (AUROC). This is calculated from the binary ground truth labels (normal or anomaly) and the anomaly rating that is predicted by the model, which typically ranges from 0 to 1. In practice, a threshold must be defined to convert this rating to a normal or anomaly result. Finding this threshold can be challenging since it must be empirically determined from a dataset. The threshold can be biased and inaccurate if the testing set contains out of distribution images from the training set. Figure 6 shows the distribution of anomaly ratings for the normal images in the training set, and Figure 6 shows the distribution of anomaly ratings for all the images in the test set. To convert the anomaly rating to a normal or anomaly label, we take the training set and take three thresholds - the maximum, the 90th percentile, and the 80th percentile anomaly rating values. These values are used on the test dataset to determine if an image is normal or anomaly, and an accuracy percentage can be calculated when compared against the ground truth measurements. To consider the best performance of the models, we report values using the threshold that has the highest performance for each model and dataset. The performance using all thresholds is shown in Appendix A.3. For LLM based models, the model responds "yes" or "no", so the response is binary. An AUROC metric cannot be calculated for these models, but a threshold is not needed to calculate accuracy scores. In our experiment, we use accuracy as the metric to compare all models.

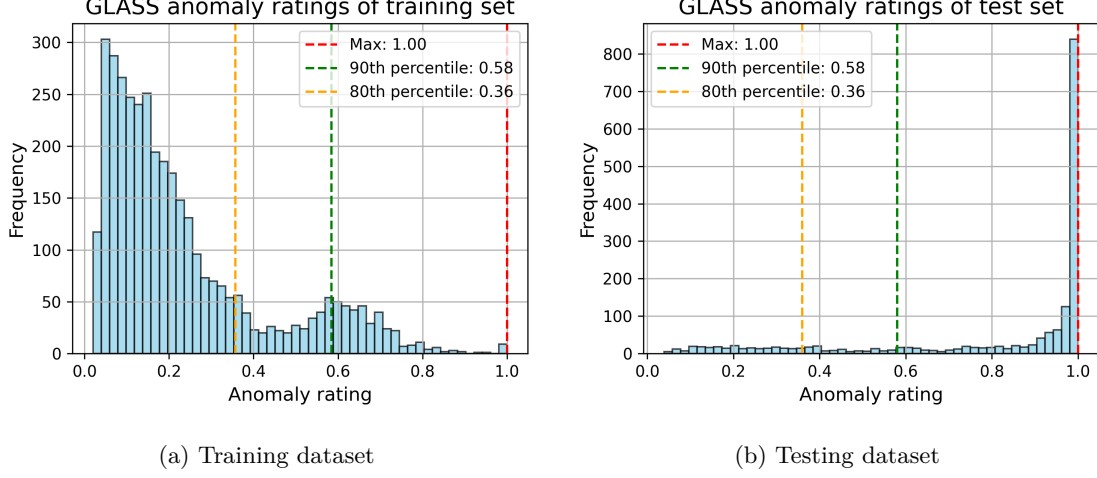

(a) Training dataset      (b) Testing dataset

Figure 6: The anomaly rating distributions in training and testing datasets for GLASS model. The different thresholds are shown as dashed vertical lines. These thresholds need to be empirically determined, and can drastically change the accuracy of the model in inference.

# 5 Results

In this section, we discuss the results on the original datasets between the SOTA models and LVLM approach. Then, we discuss the results and drop in performance for these models with modified images.

## 5.1 Results on the original datasets

The results on running each model on the original datasets are shown in Table 1 and 2. The reported AUROC and accuracy values are shown in percentages.

All three SOTA models show high performance using AUROC metrics, but drop significantly when converting to accuracy metrics for both datasets. In the MvTec-AD experiment, even when selecting the highest result between the three thresholds, GLASS drops 16.8%, INP-Former drops 14.4%, and AnomalyGPT drops 8.5%. Similarly evaluating the VisA experiment shows that GLASS drops 7.7%, INP-Former drops 5.4%, and AnomalyGPT drops 9.8%. This demonstrates the importance of setting a correct threshold when labeling normal and anomaly in an applied setting, since only using a high AUROC value to evaluate model performance can show misleading inflated performance.

In comparison, the simplest method using Qwen demonstrates a more robust performance. In the MvTec-AD dataset, the accuracy on the original images is the highest of the models at 91.8%. The superior performance of Qwen is further emphasized by the single image used for one-shot reference. While the other models have starting time costs of fine-tuning the model using the few thousand training images, Qwen can be deployed more quickly to new applications with just a single labeled normal image. The results on the VisA dataset demonstrate the limitations of Qwen. The performance of Qwen on the original dataset is lower than the other models at 73.7%.

In exploring the differences between the datasets, we hypothesize that the size of the anomaly correlates with the performance of the LVLM. There is a significant difference between the percentage of the ground truth mask percentage of the anomaly as discussed in Section 4. The average size of anomaly of the MvTec-AD dataset is 3.4% larger than that of the VisA dataset. This indicates that the anomalies in the VisA dataset may be more difficult to detect. Figure 7 plots the average mask percentage of the images binned at 50 intervals and their corresponding average accuracy for Qwen. The correlation coefficient between mask percentage and accuracy is 0.825. This demonstrates that the LVLM approach starts to fail when the anomaly becomes too small, and can predictably increase accuracy with increased anomaly size.

| Model | AUROC | Accuracy |
|---|---|---|
| GLASS | 99.9 | 83.1 |
| INP-Former | 97.7 | 83.3 |
| AnomalyGPT | 94.1 | 85.6 |
| Qwen | NA | 91.8 |

Table 1: SOTA anomaly detection models and Qwen when tested on the original MvTec-AD dataset.

| Model | AUROC | Accuracy |
|---|---|---|
| GLASS | 97.3 | 89.6 |
| INP-Former | 96.2 | 90.8 |
| AnomalyGPT | 87.4 | 77.6 |
| Qwen | NA | 73.7 |

Table 2: SOTA anomaly detection models and Qwen when tested on the original VisA dataset.

## 5.2 Results on modified datasets

The test datasets with realistic variations in images demonstrate that reported high performance may not necessarily reflect robustness of the model to changes that might occur to images in practice. All three SOTA

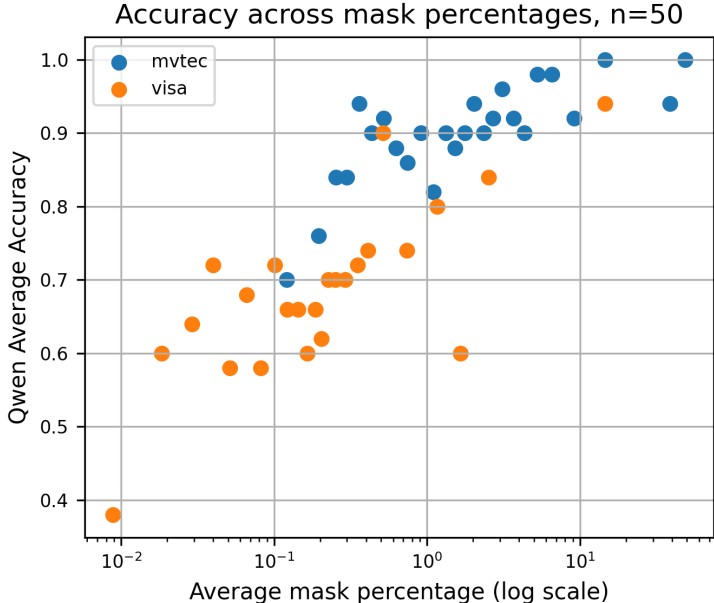

Figure 7: Average accuracy of Qwen compared to average mask percentages binned by 50 images. The average mask percentages are shown in log scale. The MvTec-AD images show overall higher average mask percentages, and correlate to higher average accuracy. The VisA images show lower average mask percentage, and correlate to lower average accuracy. The correlation coefficient is 0.825.

models have even further drop in performance when evaluated on the modified datasets, as shown in Figure 3 and 4. When selecting the threshold that yields the highest original image performance, GLASS drops an average of 7.9%, INP-Former drops an average of 4.8%, and AnomalyGPT drops an average of 10.3% in accuracies across rotation, padding, and timestamp experiments in MvTec-AD experiments. Similarly in the VisA experiments, GLASS drops an average of 26.1%, INP-Former drops an average of 14.9%, and AnomalyGPT drops an average of 11.6% across rotation, padding, and timestamp experiments. These results demonstrate that the models are not robust to changes in the images even though they do not change the normal or anomaly characteristics.

In comparison, the simplest method using Qwen demonstrates a more robust performance. In the MvTec-AD dataset, the average drop in performance across the modified images from the original performance is 1.7%, the lowest of the models. The VisA dataset, while a lower accuracy, still demonstrate the robustness across modified images with a drop in performance of 3.0%, less than the other models. While the other models may increase performance if fine-tuned on these modified images, they have starting time costs. Fine-tuning the models every time there are new testing image types is impractical and time consuming. In contrast, Qwen can be deployed more quickly to new applications with just a single labeled normal image even with new testing image types.

| Model | Original | Rotated | Padded | Timestamp | Average drop |
|:---:|:---:|:---:|:---:|:---:|:---:|
| GLASS | 83.1 | 77.1 | 73.1 | 75.3 | 7.9 |
| INP-Former | 83.3 | 67.2 | 85.6 | 82.7 | 4.8 |
| AnomalyGPT | 85.6 | 85.1 | 70.0 | 72.8 | 10.3 |
| Qwen | 91.8 | 89.4 | 89.6 | 91.3 | 1.7 |

Table 3: SOTA anomaly detection models and Qwen when tested on the modified MvTec-AD dataset.

| Model | Original | Rotated | Padded | Timestamp | Average drop |
|---|---|---|---|---|---|
| GLASS | 89.6 | 57.2 | 56.6 | 76.8 | 26.1 |
| INP-Former | 90.8 | 56.6 | 80.4 | 90.7 | 14.9 |
| AnomalyGPT | 77.6 | 69.2 | 74.2 | 55.3 | 11.6 |
| Qwen | 73.7 | 67.3 | 71.4 | 73.3 | 3.0 |

Table 4: SOTA anomaly detection models and Qwen when tested on the modified VisA dataset.

## 5.3 Ablation studies

We have demonstrated the performance of the Qwen model with one reference image and prompt. We perform ablation studies to show the efficacy of the reference image and prompt. Results for MvTec-AD is shown in Table 5.

| Qwen Model | Accuracy |
|---|---|
| Oneshot + specific prompts | 91.8 |
| Oneshot | 90.0 |
| Zeroshot + description | 76.1 |
| Zeroshot | 69.7 |

Table 5: Ablation experiment results on MvTec-AD dataset

The oneshot and specific prompts shows the best performance. When taking away the specific prompts and only using the reference image and anomaly question, the performance slightly goes down. This demonstrates that some images need the description of variances in the training set, and a single reference image alone cannot cover all variations of a normal image. The zeroshot approach with a prompt description of the anomaly object with no reference image has much lower performance, demonstrating that the single reference image is important in communicating the anomaly description. Lastly, with no prompt description and no reference image, the performance is much lower, demonstrating that language is also important in anomaly detection with LVLMs.

## 6 Future Work and Limitations

The presented LVLM approach to IAD has a few limitations. First, there is no localization of the anomaly, so pixel-level accuracy cannot be calculated. However, in practice, detecting the presence of an anomaly is more important before localizing the anomaly. In addition, the text response of the LVLM describes the anomaly, as shown in Figure 8.

Another limitation is the GPU size needed to run the LVLM and the processing time. We ran our experiments on two Nvidia A100 40GiB GPUs, and each query image took an average of 4.74 seconds to run. While some applications may not fit these requirements, the time and labor needed to accumulate a large training set and train a specialized model can be more costly, and a model that is immediately deployable may be more favorable. This limitation may be resolved with distilling the LVLM, which can boost the speed of each runs and requires less GPU to deploy.

Lastly, it is important to note that various techniques have been shown in literature to boost the performance of LVLMs. Fine-tuning, cropping, and chain-of-thought prompting are techniques that have been demonstrated to improve the performance of LVLMs (Sahoo et al., 2024). In our experiments, the specific prompts used are generic and developed quickly without iteration, but more prompt tuning can further improve the performance.

One future direction to consider is defining more detailed tasks for standard IAD benchmarks. The standard IAD datasets such as MvTec-AD and VisA only contain normal and anomaly images, and the normal images in the training set is used to define the IAD task. Because these do not contain any descriptions about what qualifies as an anomaly, the user is left to only use intuition. Figure 9 shows examples of normal and

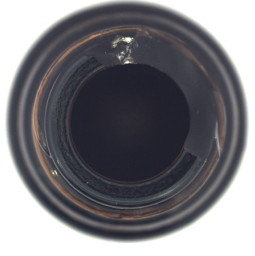

Yes. The second image shows a noticeable anomaly compared to the first image. In the second image, there is a visible crack or break in the glass near the top edge, which is not present in the first image. This crack indicates damage to the object, which is the anomaly.

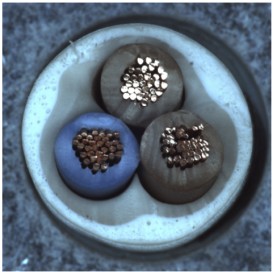

Yes. In the second image, the yellow cable with green stripes from the first image has been replaced by a brown cable. This is the anomaly.

Figure 8: An example of responses for the bottle and cable images in the MvTec-AD dataset. The text describes that the bottle anomaly is located near the top edge, and the cable anomaly has the wrong cable where the yellow cable should be.

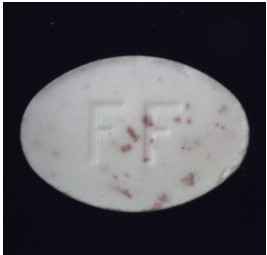 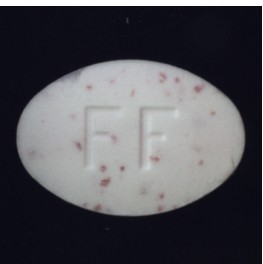 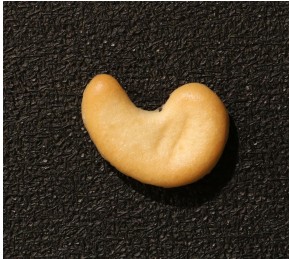 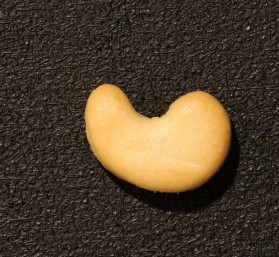

(a) Normal example of pill, but there is a noticeable chip on the right of the pill.

(b) Anomaly example of a pill, with a similar sized chip on the top of the pill.

(c) Normal example of a cashew, but there are deep scratches.

(d) Anomaly example of a cashew, with a scratch that seems less prominent than the normal image in (c).

Figure 9: Normal and anomaly images in both datasets. The pill from MvTec-AD has a similar sized chip on both normal and anomaly images (a) and (b). The cashew from VisA has a scratch on both normal and anomaly images (c) and (d). This shows that the datasets relying on using only images to define normality is ambiguous. The user of the datasets is left unclear as to what extent these imperfections render the image an anomaly.

anomaly labeled images in the MvTec-AD and VisA test datasets that are ambiguously defined. Figure 9a has a noticeable chip on the right side of the image, but is labeled normal. Figure 9b has a similar chip on the top side, but is labeled as anomaly. Similarly, Figure 9c shows a scratch in the center, but is labeled normal. Figure 9d has a seemingly smaller scratch near the center but is labeled anomaly. This lack of description makes IAD datasets ambiguous because it is unclear which type of chip on the pill or which scratch on the cashew is an anomaly. A description in language can clarify to what extent a characteristic such as these scratches becomes classified as an anomaly.

## 7 Conclusion

Our results highlight the robustness and flexibility of using LVLM for IAD with variations in images that can occur in real applications. With a simple approach of one-shot prompting, the LVLM beats SOTA models on the MvTec-AD dataset. The LVLM approach is more generalizable with the lowest drop in performance with modified images when compared to SOTA models. However, the LVLM reaches limitations when the anomaly size becomes small, as shown by the lower performance on the VisA dataset. We also demonstrate that the nature of IAD should not rely only on images, since there may be wide variations of a normal image, and missed variations in the training set may be classified as an anomaly. By combining a single reference image and language, the LVLM approach is more flexible to changes in the images in practice. It can also be more quickly deployed without obtaining a large training dataset and fine-tuning. Therefore, LVLMs show promising potential as a faster and scalable approach to IAD.

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

# A Appendix

## A.1 Details of running Qwen

To explore the performance of these models on IAD tasks, we evaluate the Qwen2-VL-72B-Instruct-AWQ model on IAD datasets (Wang et al., 2024a). We run this model on two NVIDIA A100 GPUs with 40GB memory with vLLM to speed up the processing time. We used minimum and maximum pixels 768*28*28 and 1024*28*28 respectively, max_num_seqs is 5, max_model_len is 7000, temperature is 0.3, and max_tokens is 1024.

### A.2 Additional prompts

The additional prompts used for some image classes are shown in Table 6 and 7. Most image classes did not need these prompts, because the single reference image was enough. However, some classes had variations that were considered normal, such as different colors of toothbrush bristles, that were not conveyed by the single reference image.

| Image Class | Specific Prompt for Original, Padding, and Timestamp images | Specific Prompt for Rotating images |
|---|---|---|
| bottle | NA | NA |
| cable | NA | Ignore differences in the position of the objects. |
| capsule | Ignore differences on the "actavis" text on the black surface. | Ignore differences on the "actavis" text on the black surface. |
| carpet | NA | NA |
| grid | Ignore differences in color tones. | Ignore differences in color tones. |
| hazelnut | NA | NA |
| leather | NA | NA |
| metal nut | NA | NA |
| pill | NA | NA |
| screw | NA | NA |
| tile | NA | NA |
| toothbrush | Ignore the color of the bristles for an anomaly. | Ignore the color of the bristles for an anomaly. Ignore the angle of the toothbrush head for an anomaly. |
| transistor | NA | NA |
| wood | NA | Ignore differences in vertical or horizontal grain pattern for an anomaly. |
| zipper | NA | NA |

Table 6: Specific prompts used for the MvTec-AD dataset. Most images did not require any specific prompt, and used the generic prompt shown in Figure 3.

| Image Class | Specific Prompt for Original, Padding, and Timestamp images | Specific Prompt for Rotating images |
|---|---|---|
| candle | NA | NA |
| capsules | NA | NA |
| cashew | NA | NA |
| chewinggum | NA | NA |
| fryum | NA | NA |
| macaroni1 | NA | NA |
| macaroni2 | Ignore the orientation of the objects. | Ignore the orientation of the objects. |
| pcb1 | NA | NA |
| pcb2 | NA | NA |
| pcb3 | NA | NA |
| pcb4 | NA | NA |
| pipe fryum | NA | NA |

Table 7: Specific prompts used for the VisA dataset. Most images did not require any specific prompt, and used the generic prompt shown in Figure 3.

### A.3 Results from different thresholds

| Dataset | Accuracy |
|---|---|
| Original MvTec-AD | 57.8/83.1/79.3 |
| Rotated MvTec-AD | 56.1/77.1/76.5 |
| Padded MvTec-AD | 57.0/73.1/73.0 |
| Timestamp MvTec-AD | 58.6/75.3/72.8 |
| Original VisA | 89.6/87.1/81.3 |
| Rotated VisA | 57.2/56.6/56.6 |
| Padded VisA | 56.6/56.6/56.6 |
| Timestamp VisA | 76.8/74.2/70.6 |

Table 8: GLASS performance with max threshold/90th percentile threshold/80th percentile threshold.

| Dataset | Accuracy |
|---|---|
| Original MvTec-AD | 83.3/82.8/78.9 |
| Rotated MvTec-AD | 67.2/74.6/74.6 |
| Padded MvTec-AD | 85.6/76.8/75.5 |
| Timestamp MvTec-AD | 82.7/85.0/81.1 |
| Original VisA | 90.8/83.3/76.6 |
| Rotated VisA | 56.6/56.6/56.6 |
| Padded VisA | 80.4/71.6/65.0 |
| Timestamp VisA | 90.7/82.9/76.0 |

Table 9: INP-Former performance with max threshold/90th percentile threshold/80th percentile threshold.

