# OpenReview forum: "Rethinking Industrial Anomaly Detection in the Era of Large Vision-Language Models"
_TMLR — Withdrawn by Authors_

### Review · Reviewer_VFnM · 2025-09-22

**Summary Of Contributions:**

This paper presents an approach to industrial anomaly detection (IAD) through the use of pretrained large vision-language models (LVLMs). The authors argue that the traditional framing of treating everything outside of a curated training set as “anomalous” may lead to false positives, which they hypothesize might be mitigated with LVLMs. Experiments across two IAD benchmark datasets show that prompting a publicly available LVLM gave mixed results, outperforming baselines on one dataset but underperforming on another. However, the LVLM-based approach appeared to be more robust to transformations like rotation, timestamping, etc. than other methods.

**Audience:**

Yes

**Audience Explanation:**

The problem setting should be of interest to a subset of readers. However, prior studies have far more comprehensively attempted to measure the utility of large vision-language models for industrial anomaly detection.

**Broader Impact Concerns:**

A Broader Impact statement may be necessary for this application. Perhaps some reference to discussion of ethics related to automating jobs and tasks like this may be appropriate. E.g., who is to blame for errors and what are the broader costs of potentially putting human workers out of a job? Or rather, is there still a role for humans to play in this process?

**Claims And Evidence:**

No

**Claims Explanation:**

**Strengths**:
- The problem setting is interesting and is likely to have broad research and commercial interest/value.

**Weaknesses**:
- The unique contribution of this paper needs to be made clearer. There is no technical novelty, so I presume this is not meant to be a methodological contribution. However, if this is more of a “benchmark paper” (demonstrating the ability of LVLMs for IAD), then there is much work to be done in strengthening this benchmark.
- Related to the above point, experimental validation can be expanded to more datasets and, minimally, to more baseline methods and LVLMs.
- Discussion of prior work is lacking. This is a very well-studied problem, so the burden to differentiate this work from prior studies is quite high. I can find dozens of examples of papers using vision-language models for IAD [1-6].
- Writing can be improved and generally formalized throughout. In my opinion, this reads more like a technical report than a research paper.

**References**

[1] Gu, Zhaopeng, et al. "Anomalygpt: Detecting industrial anomalies using large vision-language models." Proceedings of the AAAI conference on artificial intelligence. Vol. 38. No. 3. 2024.

[2] Li, Jiaqi, Shuhuan Wen, and Bin Fang. "G-Anomaly: A pyramid graph Transformer-based Vision-Language model for general industrial anomaly detection." IEEE Transactions on Automation Science and Engineering (2025).

[3] Chen, Zhiling, et al. "Can multimodal large language models be guided to improve industrial anomaly detection?." arXiv preprint arXiv:2501.15795 (2025).

[4] Jiang, Xi, et al. "Mmad: A comprehensive benchmark for multimodal large language models in industrial anomaly detection." arXiv preprint arXiv:2410.09453 (2024).

[5] Deng, Huilin, et al. "Vmad: Visual-enhanced multimodal large language model for zero-shot anomaly detection." IEEE Transactions on Automation Science and Engineering (2025).

[6] Cao, Yunkang, et al. "Personalizing vision-language models with hybrid prompts for zero-shot anomaly detection." IEEE Transactions on Cybernetics (2025).

**Requested Changes:**

**Main feedback**
- Clarify the positioning of this paper. Is it a benchmark? If so, what distinguishes it from prior work? [4]
- Expand the discussion of prior work to include the wealth of vision-language approaches (for a short list, see [1-6]).
- Expand experiments to include more LVLMs. Why is only Qwen benchmarked?
- If possible, expand experiments to more datasets and tasks (not necessarily binary “anomaly” vs. “normal”). Transformations can be combined and made more realistic, and use of a real-world dataset may add value as well.
- Why do the authors think that Qwen underperformed on one dataset and overperformed on another? Further, why was it more robust to transformations than the other methods?


**Minor feedback**
- Abstract: Remove comma “state-of-the-art models, and offer”
- Abstract: What is meant by “small” abnormalities?
- Introduction: Provide evidence for this claim if possible, “Uncommon characteristics of images that are not included in the training set may be considered an anomaly, even when the image is normal.”
- Related Work: Informal phrasing can be incorporated into sentences around it, “Some methods include [xyz]”.
- Sec 3: What determined which classes needed more specialized prompts?
- Generally, phrases like “embedding based” and “texture based” should be written “embedding-based” and “texture-based”
- Sec 4.3: Change “Augmented testset” -> “Augmented test set”
- Sec 4.4: Awkward phrasing. Perhaps change “The SOTA models standard of evaluation metric” -> “The standard metric used to evaluate SOTA models is”
- Sec 5.1: Informal phrasing, “The results on running each model on the original datasets are shown…”
- Sec 5.1: AUROC and accuracy cannot be directly compared—they are completely different metrics.

---

### Review · Reviewer_G3WA · 2025-09-25

**Summary Of Contributions:**

This paper presents an empirical study on the ability of various large vision-language models (LVLMs) to detect industrial abnormalities. The study is conducted in a one-shot setting, where a reference image of normal conditions is provided as context. Testing images are then input into the LVLMs along with a custom text prompt that inquires about the presence of abnormalities and requests an anomaly rating score. A threshold, determined using the training set, is applied to convert the anomaly rating score into final classification decisions.

The study utilizes two publicly available datasets and evaluates four open-source vision-language models. Ablation studies are performed in a zero-shot setting and explore variations in prompts. Additionally, an ablation study examines performance under different modifications to the testing images.

Pros:
1. The paper provides a valuable benchmark and profiling report on the usability of vision-language models (VLMs) for a specific application challenge. This contribution could assist the community in gaining a better understanding of VLM capabilities from various perspectives.
2. The concept of evaluating modified / augmented test cases is interesting. Given that large language models (LLMs) are known to overfit public dataset testing sets, the paper's approach to evaluating each method could offer valuable insights for future research.

Cons:
1. To enhance the manuscript and make it a more comprehensive and insightful scientific report, a few additional elements could be considered. Please refer to following explanations.

**Additional Comments:**

None

**Audience:**

Yes

**Audience Explanation:**

An comprehensive report today could be beneficial for the community to understand the potential / limitations of LLM / LVLM on various domain. Although there is no algorithm / theorem proposed in this paper, it could also potentially be interesting to TMLR readers

**Broader Impact Concerns:**

Not relevant

**Claims And Evidence:**

No

**Claims Explanation:**

1. The AUC results seem to present an interesting contrast to the data distribution shown in Figure 6. Considering the thresholding method employed in the paper and the noticeable difference between the training and test set distributions, one might anticipate a higher false positive rate for GLASS rather than an almost perfect AUC. It might be beneficial for the authors to also provide the average precision score to offer a more comprehensive view of the precision and recall for each LVLM.

2. There are several other open-source LVLMs, such as LLaVA, Pixtral, and NVLM, that are not included in the study. Since the paper aims to benchmark the performance of existing methods, it could be advantageous to include a broader range of models to enhance the comprehensiveness of the study.

3. The report currently does not include information on running costs, which might help readers evaluate the results within a well-defined experimental framework.

4. Additional ablation studies could be valuable in helping readers better understand the results. For instance, the authors might consider exploring few-shot settings, SFT settings, and different sizes of each baseline to provide a more thorough analysis.

5. Including examples of failure cases could be quite helpful. They can offer readers deeper insights into the limitations of LVLMs.

6. It is intriguing to note that rotation can lead to significant performance deterioration, even with modern DNN architectures. This observation raises some questions about the uncertainty and robustness of the results. It might be worthwhile for the authors to conduct multiple queries on each input and report the standard deviation of the scores to provide additional clarity.

7. The table appears to be missing the number for the most important baseline, Qwen. Could this be a typographical error, or is there a specific reason for its omission?

**Requested Changes:**

Please refer to 'explain your answer above' section

---

### Review · Reviewer_4trP · 2025-10-10

**Summary Of Contributions:**

This paper proposes an LVLM-based approach to industrial anomaly detection(IAD), aiming to address the time and cost burdens of collecting large amounts of normal data. The method bypasses training by relying on a pretrained LVLM with strong generalization ability. Given a single normal reference image and a test image, the model is prompted to determine whether the test image is anomal. By adjusting the prompt, the approach can also be tailored to detect only specific types of anomal images that either meet or fail to meet certain conditions.

**Audience:**

Yes

**Audience Explanation:**

1. The idea of leveraging LVLMs to simplify IAD is novel and appealing. The one-shot design eliminates training time, and with only a single reference image required, both data preparation and computation are significantly reduced compared to conventional methods.
2. The prompt design around a reference image is intuitive, and the use of an open-source model like Qwen to demonstrate performance is commendable. These choices enhance reproducibility, making it straightforward for others to replicate the results.

**Broader Impact Concerns:**

The work raises no immediate ethical red flags. However, it would be useful for the authors to briefly discuss the implications of deploying LVLM-based anomaly detection in safety-critical industrial environments.

**Claims And Evidence:**

No

**Claims Explanation:**

While the submission presents experimental results on two benchmark datasets (MvTec-AD and VisA), the evidence may not be fully convincing to support the broader claims of the paper. The experiments are restricted to two datasets, and the reliance on accuracy as the main evaluation metric does not sufficiently reflect the characteristics of imbalanced anomaly detection tasks. Consequently, the results only partially demonstrate the robustness and generalization ability of the proposed approach.

**Requested Changes:**

1. While the idea of leveraging LVLMs is interesting, the methodological contribution is not clear compared to existing anomaly detection research. The approach often looks like a prompt-engineering variant rather than a fundamentally new methodology.

2. The experiments are confined to MvTec-AD and VisA. This raises the concern that the results may be overly tuned to the characteristics of these datasets. To validate the generalization ability of LVLMs, it would be important to test on a wider range of industrial data.

3. The comparison between AUROC and accuracy is a useful observation, but additional metrics are necessary. In anomaly detection, AUROC is generally preferred over accuracy because anomaly data are very rare, and the cost of misclassifying an anomaly (false negative) is much higher than the cost of a false positive. For example, in cancer prediction, correctly identifying a patient with cancer (high TPR) is far more critical than misclassifying a healthy person as positive (FPR). For this reason, AUC, F1-score, and other imbalance-sensitive metrics are standard in highly imbalanced datasets. It is risky to rely only on accuracy. So alternative evaluation metrics beyond accuracy are necessary.

4, The discussion on threshold is sharp, but the analysis of why models with high AUROC can still be highly threshold-dependent is necessary. One plausible reason is that some anomal data may cluster tightly, causing TPR to rise sharply around certain thresholds. In such cases, whether the chosen threshold includes or excludes this cluster can have a large impact on accuracy. More exploration of these underlying causes would strengthen the argument.

5. Using a single reference image is computationally efficient, but it also raises concerns about robustness. Since the reference image can naturally vary in conditions such as angle, lighting, or background, it would be important to evaluate how sensitive the results are to such variations. Experiments with multiple reference images would provide stronger evidence of robustness.

---

### Note · Authors · 2025-10-29

**Comment:**

We sincerely thank the reviewers for their thoughtful and constructive comments. We greatly appreciate the time and effort taken to evaluate our work, and we recognize that the suggestions provided will significantly enhance the quality and impact of our paper.

Several of the recommended experiments are substantial and require additional time to complete thoroughly. We have already begun conducting some of these studies and have obtained interesting preliminary findings. However, the full set of experiments is not yet finished.

Given this, we plan to continue our investigation and incorporate the reviewers’ valuable feedback into a more comprehensive future version of the paper. We appreciate the suggestions of the reviewers because the new experiments have obtained interesting preliminary findings. We look forward to submitting an improved manuscript as a new submission (with this submission linked as a previous submission url) once the additional experiments and analyses are complete.

**Withdrawal Confirmation:**

I have read and agree with the venue's withdrawal policy on behalf of myself and my co-authors.